# Assessment of dream-related aspects and beliefs in a large cohort of French students using a validated French version of the Mannheim Dream questionnaire

Alice Napias[1], Emilie Denechere[2], Willy Mayo[1,3], Imad Ghorayeb[1,2,3]*

**1** Institut de Neurosciences Cognitives et Intégratives d'Aquitaine - UMR 5287, Université de Bordeaux, Bordeaux, France, **2** Department of Clinical Neurophysiology, University Hospital of Bordeaux, Bordeaux, France, **3** CNRS, Institut de Neurosciences Cognitives et Intégratives d'Aquitaine UMR 5287, Bordeaux, France

* imad.ghorayeb@u-bordeaux.fr

**Data Availability Statement:** All relevant data are within the manuscript and its Supporting information files.

## Abstract

Focusing on a specific population when studying dream characteristics can shed light on underlying mechanisms and correlates of dreaming. The aim of this study is to establish a clearer description of specific dream aspects and beliefs in a large cohort of students using a validated questionnaire, and to further investigate the role of sociodemographic variables such as age, gender and field of study. Participants were 1137 students aged from 18 to 34 (mean age: 22.2) who responded to an online version of the questionnaire. Our results showed a difference between humanities and science students, and a differential effect of gender on dream variables. Our results are discussed in light of previous investigations using the same questionnaire or focusing on the same population.

## Introduction

Dreaming is a subjective psychological state that is highly correlated with the occurrence of rapid eye movement (REM) sleep [1]. The dream experience is a sequence of sensations, images, emotions and thoughts that can be recalled and expressed during wakefulness, especially when this experience is accompanied by vivid perceptual and emotional content, such as a high degree of bizarreness [2]. Since the discovery of REM sleep, scientific knowledge on the relationship between dreaming and physiological measures including brain activity has accumulated [3–5]. However, whether dreams serve any specific function or are just a byproduct of brain activity while we sleep remains a matter of debate. Several theories have been formulated over past decades that attempt to explain dreaming, including those that assign specific functions to this process (as opposed to REM sleep) that include its potential role in emotional regulation [6,7]. Due to its subjective nature, dream experience is inaccessible to objective instruments but it can be indirectly assessed by self-report measures such as questionnaires or diaries. Collecting information on dream experiences and beliefs from large cohorts is therefore an important resource for investigating the role of sociodemographic and cultural variables in the dreaming process,

**Funding:** The authors received no specific funding for this work.

**Competing interests:** The authors have declared that no competing interests exist.

dream perception and dream interpretation. In particular, such epistemological studies should help advance understanding of the extent to which sociodemographic factors are correlated with dreaming, thereby allowing theories to integrate these variables in their formulation of dream functions. Through large-scale questionnaire studies, several variables such as age, gender, and personality have been linked to dream characteristics. General findings report that young adults, women and people who are more open to experiences are more likely to recall dreams compared to older adults, men and people less open to experiences [8–12].

University students constitute a population that may be at particular risk to experience sleep disturbances [13], and this fact might interfere their ability to remember their dreams. Although the student population is more homogenous than the general population because of more common lifestyles and a limited age range, differences between subpopulations of students can be identified. When compared to medical students, for example, humanities students demonstrate higher scores on the personality traits of extroversion, emotional stability and openness to experience [14]. The ability to remember dreams could therefore differ between students focusing on humanities as compared to students enrolled in the medical or technological sciences. However, studies conducted in the goal of evaluating the implication of sociodemographic variables on dream variables are scarce in the student population. In France, only one study to our knowledge has reported sleep and dream habits in a sample of French college students using an online non-validated questionnaire [15]. In their sample, age and gender were both related to dream recall frequency, but the authors did not observe differences in dream recall frequency between students in the humanities and sciences. They also did not evaluate attitude towards dreams, such as the extent to which people focus on their dreams and think over them, which could help explain the relationships between dreaming experience and sociodemographic factors or academic domain. To aid understanding of such results, use of the same scales of frequency by different research groups would allow for a more precise comparison of findings across different samples.

The Mannheim Dream questionnaire (MADRE) is a comprehensive instrument that evaluates dream recall frequency, emotional aspects of dreams, nightmares, lucid dreaming, attitude towards dreams, effects of dreams on waking life and dream literature reading, with high reliability [16]. Our team previously validated the French version of the MADRE and demonstrated high test-retest reliability [17]. This questionnaire was also validated in German [16] and translated into Persian [18,19] and Italian [20], and a first version was also edited in French [21]. This questionnaire has been used to assess between-person differences in dream variables based on sociodemographic characteristics such as age and gender, but also based on the dream recall frequency of the respondents. In the present survey, the validated French version of the MADRE was submitted to a large cohort of French students drawn from a larger web-based study (the i-Share project) including approximately 20.000 students.

The goal of this study is to focus on the homogeneous student population and its specificity to account for differences in dream recall frequency and other secondary dream variables. Our specific aims are to: (i) provide a clearer description of specific aspects of dreaming in this population; (ii) extend our previous findings regarding the relationships between dreaming and sociodemographic variables; and (iii) interpret our results relative to the prevailing literature using the same questionnaire [16,17,20,21].

## Materials and methods

This study was approved by the ethical committee "*Comité de Protection des Personnes pour la région de l'Est* 1" (CPP EST 1) (ID RCB N˚ 2017-A03360-53, CPP EST 1 N˚ 2018/03). All of our data are available online in the S1 Dataset and S1–S4 Tables section of the article.

## Research questionnaire

For the purpose of the study, we used the validated French version of the MADRE [17] in an online format. The link to the questionnaire was posted on the i-Share project website and the information was broadcast through the i-Share and the University of Bordeaux social networks, as well as the homepage of the University Hospital of Bordeaux website between May 26[th] 2018 and June 8[th] 2018.

The MADRE questionnaire regroups frequency scales and questions regarding different topics concerning dreams. The French version used in this study is available online [17]. A first scale evaluates the dream recall frequency (on a seven-point scale coded as 0 = Never, 1 = Less than once a month, 2 = About once a month, 3 = About 2 to 3 times a month, 4 = About once a week, 5 = Several times a week, 6 = Almost every morning). Emotional characteristics of dreams are also assessed, with intensity (on five-point scale coded as 0 = Not at all intense, 1 = Not that intense, 2 = Somewhat intense, 3 = Quite, 4 = Very intense) and overall tone (on five-point scale coded as -2 = Very negative, -1 = Somewhat negative, 0 = Neutral, +1 = Somewhat positive, +2 = Very positive). Nightmare frequencies are assessed through two frequency scales (current and childhood occurrence on eight-point scales coded as 0 = Never, 1 = Less than once a month, 2 = About once a year, 3 = About 2 to 4 times a year, 4 = About once a month, 5 = About 2 to 3 times a month, 6 = About once a week, 7 = Several times a week). Nightmare characteristics are measured with a five-point scale of disturbance (coded as 0 = Not at all distressing, 1 = Not that distressing, 2 = Somewhat distressing, 3 = Quite distressing, 4 = Very distressing), a question evaluating whether the participants experienced recurrent nightmares related to waking-life situations (Yes/No) and a question about the proportion of nightmares that are recurrent (as a percentage). Lucid dream experiences are evaluated with one eight-point frequency scale and one question about the age of first occurrence (approximate or imprecise responses were coded as NA or X and were not included in the analyses). Several eight-point frequency scales evaluate behaviors such as dream sharing; recording dreams; problem-solving dreams; creative dreams; dreams affecting daytime mood; and déjà-vu experiences. Another dimension assessed the tendency of reading about dreams (0 = No, 1 = Once or Twice, 2 = Several times) and the evaluation of usefulness of this reading (on a five-point scale coded as 0 = Not at all, 1 = Not that much, 2 = Somewhat, 3 = Quite, 4 = Very much). Finally, attitude towards dreams is estimated through 6 five-point scales of participant agreement with statements (coded as 0 = Not at all, 1 = Not that much, 2 = Partly, 3 = Somewhat, 4 = Totally), and that showed high internal consistency in previous studies [16,17] as well as in our present sample (McDonald's Omega = .846). A confirmatory factor analysis conducted after verifying conditions for its application (KMO = .85; Bartlett test's $Chi^2$ = 2527.4, df = 15, p < .05) showed a unique factor regrouping all of the items. This factor, corrected to fit the non-normality of the data, regrouped all of the items as well as covariation between two different items, with correct SRMR < .08, RMSEA < .06 and CFI >.90 [22]. Using the same five-point scales of agreement, two items measured the personal meaning of one's own dream and the impression that dreams provide insight or hints for waking life.

## Study population

**Recruitment.** Study participants were unpaid students 18 years of age or over and enrolled in the ongoing internet-based Students Health Research Enterprise (i-Share) project, a prospective population-based e-cohort study of students in French-speaking universities and institutions of higher education. To be eligible to participate in the i-Share project cohort, the student had to be able to read and understand French and provide informed consent for participation. The i-Share project was approved by the "*Commission Nationale de l'Informatique*

*et des Libertés*" (CNIL—National Commission of Informatics and Liberties) [DR-2013-019]. Data accumulated as part of this study came from students based in Bordeaux. Participants were informed about the purpose and aims of the study via social media networks (http://www.i-share.fr) and anonymously completed the self-administered questionnaire online. The questionnaire was also filled by adults that were not students and minor students, their data were removed from our dataset.

**Sample description.**   A total of 1137 respondents completed the online questionnaire. Of the total cohort, 923 were women (81%) and its mean age was 22.2 ± 2.58 years (min = 18, max = 34). Half of the sample (N = 555, 49%) were completing studies in the humanities and the other half in scientific disciplines (N = 582, 51%). Women and men are equally distributed between the two fields of study, with 50% (N = 466) of women and 54% (N = 116) of men studying scientific disciplines.

## Statistics

All statistical analyses were carried out using R 3.5.1 software. In line with Schredl et al. [16], ordinal scales were treated with ordinal regressions to study the correlations between sociodemographic variables (age, gender, field of study) and different dream characteristics such as dream recall frequency or nightmare frequency. In the case of numeric variable such as the percentage of recurring nightmares or age of first lucid dream, linear regressions were used. We conducted a confirmatory factorial analysis to verify the presence of one factor related to attitude towards dreams.

## Results

### Dream recall frequency

The distribution of dream recall frequency is shown in Table 1. In our sample, half of the sample reported recalling dreams at a frequency spanning several times a week to almost every morning, with the most frequent answer being "several times a week" (38.26%). Regarding sociodemographic variables, women reported significantly more dream recall than men ($\beta$ = 0.474, $Chi^2$ = 11.80, p = .0006). Within this student cohort (18 to 34 years old), age was not related to dream recall frequency ($\beta$ = -.0154, $Chi^2$ = .5454, p = .4602). Dream recall frequency was significantly associated with field of study, with humanities students reporting more dream recall than science students ($\beta$ = -0.2469, $Chi^2$ = 5.32, p = .021). With gender considered as a covariate, this relationship remained significant ($\beta$ = -0.2435, $Chi^2$ = 16.96, p = .023). When genders were considered separately, women did not report significantly more or less

**Table 1.  Dream recall frequency distribution of the total sample stratified by gender.**

|  | Gender | | | | Total population (N = 1137) | |
|---|---|---|---|---|---|---|
|  | Women (n = 923) | | Men (n = 214) | | | |
| *Categorial Variable* | n | % | n | % | n | % |
| **Dream recall frequency** | | | | | | |
| Almost every morning | 181 | 19,61 | 34 | 15,89 | 215 | 18,91 |
| Several times a week | 369 | 39,98 | 66 | 30,84 | 435 | 38,26 |
| About once a week | 187 | 20,26 | 50 | 23,36 | 237 | 20,84 |
| Two to three times a month | 113 | 12,24 | 36 | 16,82 | 149 | 13,10 |
| About once a month | 40 | 4,33 | 13 | 6,07 | 53 | 4,66 |
| Less than once a month | 24 | 2,60 | 9 | 4,21 | 33 | 2,90 |
| Never | 9 | 0,98 | 6 | 2,80 | 15 | 1,32 |

dream recall depending on their field of study ($\beta = -0.096$, $Chi^2 = .65$, p = .419), whereas men studying in the humanities reported significantly more dream recall than those studying in scientific disciplines ($\beta = -0.8636$, $Chi^2 = 12,12$, p = .0005). Looking at the median of each subgroup, men in the sciences appear to report less dream recall than the other subgroups.

## Emotional dimensions of dream experience

Concerning emotional intensity, half of the sample reported dreams as quite or very intense, with "quite intense" being the most given answer (45.56%). As for emotional tone, the most frequent answer was "somewhat negative" (35.71%), with the median position corresponding to "Neutral" tonality. In relation to sociodemographic variables, age was unassociated with the intensity of dreams ($\beta = -.0011$, $Chi^2 = .003$, p = .958), or with their emotional tone ($\beta = -.0209$, $Chi^2 = .971$, p = .359). Gender showed significant associations with both variables. For emotional intensity, women tended to report more intense dreams ($\beta = .3034$, $Chi^2 = 4.525$, p = .0334), and also tended to experience more negative dreams ($\beta = -.406$, $Chi^2 = 8.92$, p = .0028). Considering field of study, humanities students reported more intense dreams ($\beta = -0.265$, $Chi^2 = 5.79$, p = .016), and more negative dream tone overall ($\beta = .469$, $Chi^2 = 18.93$, p = < .0001) compared to science students.

## Nightmares

Half of the sample reported currently having nightmares at least once a month, while the frequent answer was "two to three times a month" (21.64%). In childhood, half of the students also reported remembering having nightmares at least once a month, with the most response being "two to three times a month" (22.87%). The correlation between current nightmare frequency and nightmare frequency during childhood was significantly and positive, r (1135) = .438, p < .0001. Comparing the two distributions, nightmares overall were more frequent during childhood than currently (Wilcoxon W = 156052, p < .0001). Out of the 1089 (95.77%) participants reporting experiencing nightmares, half of the sample experienced their nightmares as at least somewhat distressing, with "somewhat distressing" (31.31%) being the most frequent response. They reported that 38.86% of their nightmares were recurrent, and a total of 40.40% of students experiencing recurrent nightmares reported that their content was related to waking-life situations.

Regarding sociodemographic variables, age was not associated with the frequency of current nightmares ($\beta = -.005$, $Chi^2 = .074$, p = .78) or their frequency during childhood ($\beta = -0.037$, $Chi^2 = 3.296$, p = .069). Compared to men, women reported significantly more frequent nightmares both currently ($\beta = 1.067$, $Chi^2 = 59.98$, p = < .0001) and during childhood ($\beta = .319$, $Chi^2 = 5.45$, p = .019). Field of study was significantly associated with current nightmare frequency, with humanities students reporting more frequent nightmares than science students ($\beta = -.582$, $Chi^2 = 30.78$, p = < .0001), but no association was observed for childhood nightmare frequency ($\beta = -155$, $Chi^2 = 2.19$, p = .138). Regarding the degree to which current nightmares (n = 1089) were disturbing, no association was observed for age ($\beta = -.004$, $Chi^2 = .044$, p = .83), but women reported significantly more disturbing nightmares than men ($\beta = .831$, $Chi^2 = 33.08$, p = < .0001). Field of study was unrelated to disturbance ratings ($\beta = -.195$, $Chi^2 = 3.233$, p = .072).

## Lucid dream experiences

Lucid dreaming has been experienced at least once by 75.28% (n = 856) of our sample. Among them, 47.75% (n = 543) were able to estimate their age when they first experienced a lucid dream. Mean age of first lucid dream experience was 13.5 ± 3.95 years old. For the total

sample, regression analyses showed that age was not correlated with the frequency of lucid dream experience (β = .0181, Chi$^2$ = .8302, p = .362), but women experienced lucid dreams more frequently than men (β = .3148, Chi$^2$ = 5.57, p = .0183), and students in the humanities reported lucid dreams more frequently than science students (β = -.2201, Chi$^2$ = 4.46, p = .0347).

## Attitude towards dreams

An attitude towards dreams score was computed as the average of six items, with a mean of 2.11 ± 1.03 in our sample. Regarding its relationship with sociodemographic variables, age showed a significative negative relationship with attitude towards dreams (β = -.029, t = -2.439, p = .015). Gender also demonstrated a significant association with this variable (β = .23, t = 3.044, p = .0024), with women having a more positive attitude towards dreams. When both gender and age were considered in the same analysis as covariates, both remained significant predictors of the attitude towards dreams mean score (Age: β = -.026, t = -2.24, p = .025; Gender: β = -.022, t = 2.88., p = .004). When considered separately, men did not present differences depending on their age (β = -.009, t = -0.352, p = .725), but scores for women decreased with age (β = -.031, t = -2.33, p = .019). Field of study was also associated with overall attitude, with humanities students having a significantly more positive attitude than science students (β = -.188, t = -3.079, p = .002). Attitude towards dreams was strongly and positively associated with dream recall frequency (β = 0.29, t = 13.8, p = < .0001). Finally, half of the sample considered that dreams provide at least some guidance in life, and attribute meaning at least partially to their dreams.

## Effects of dreams on waking life

Half of our sample reported that they shared their dreams with others at least two to three times a month, which was also the most answered modality (21.72%). However, a large majority of the sample (77.9%) declared that they never recorded their dreams. A large percentage of our sample also reported that they never felt like dreams could affect their daytime mood (26.47%), provide creative ideas (51.10%), or help to solve problems (41.60%). Concerning déjà vu experiences, the median score was high, with half of our sample reporting that they experienced this phenomenon at least once a month and with the frequent response being "two to three times a month" (24.45%).

Table 2 presents the results of regression analysis between variables concerning dreams' effects on waking life and sociodemographic variables. Significant associations were only observed for the frequency of sharing dreams and déjà vu experiences that both diminished

**Table 2. Regression analysis for variables concerning effects of dreams on waking life.**

|  | Effect of age | | | Effect of Gender | | | Effect of Field of study | | |
|---|---|---|---|---|---|---|---|---|---|
|  | β | Chi$^2$ | p | β | Chi$^2$ | p | β | Chi$^2$ | p |
| **13. Telling dreams** | -.047 | 5.36 | .0205 | 1.319 | 89.49 | < .0001 | -.156 | 2.248 | .133 |
| **14. Recording dreams** | -.033 | 1.382 | .2397 | -.1615 | .8475 | .3573 | -.1300 | 0.8426 | .3587 |
| **15. Dreams affecting daytime mood** | -.0324 | 2.510 | .1131 | .7403 | 29.92 | < .0001 | -.1105 | 1.121 | .2898 |
| **16. Dreams providing creative ideas** | -.0061 | .0798 | .7776 | -.2024 | 2.059 | .1497 | -.1672 | 2.278 | .1312 |
| **17. Dreams solving problems** | -.0082 | .1508 | .6980 | .3213 | 5.55 | .0184 | -.209 | 3.807 | .0511 |
| **18. Déjà vu experiences** | -.1214 | 34.76 | < .0001 | .021 | .0239 | .8770 | -.262 | 6.266 | .0123 |

β = Standardized estimates.

**Table 3. Regression analysis of dream-related variables adjusting for dream recall frequency.**

| | Dream recall frequency | | |
|---|---|---|---|
| | β | Chi²/t | P |
| **12. Attitude towards dreams** | 0,29 | 13,8 | <,0001 |
| **13. Telling dreams** | 0,728 | 296,3 | <,0001 |
| **14. Recording dreams** | 0,202 | 12,84 | 0,0002 |
| **15. Dreams affecting daytime mood** | 0,508 | 145,8 | <,0001 |
| **16. Dreams providing creative ideas** | 0,344 | 61,60 | <,0001 |
| **17. Dreams solving problems** | 0,376 | 78,73 | <,0001 |
| **18. Déjà vu experiences** | 0,229 | 32,76 | <,0001 |
| **19. Reading about dreams** | 0,240 | 32,36 | <,0001 |
| **20. Useful dream literature (n = 802)** | 0,049 | 0,95 | 0,3278 |

β = Standardized estimates.

with age. Three variables were associated with gender, with women more frequently sharing their dreams, reporting dreams as affecting their daytime mood, and describing dreams as a means to solve problems. Regarding relationships to field of study, the frequency of déjà vu experience was greater for humanities students compared to science students.

## Reading about dreams

Concerning the two questions relative to reading about dreams, 70.54% (n = 802) of our sample read at least once about dreams, and half of the sample found such reading to be at least somewhat useful. The most frequent answer was nonetheless "not that useful" (38.03%). For both variables, no association was observed with age, gender or field of study.

## Analyses adjusted for dream recall frequency

Table 3 presents the regression analysis of dream variables with dream recall frequency as a covariate. Dream recall frequency appeared as an important variable to take into account to explain the variance of the different dream-related variables, except for the usefulness of dream literature. Participants reporting recalling their dreams more frequently also reported a more positive attitude towards dreams and sharing more frequently their dreams. Although 77.7% of our sample declared not recording their dreams, an association was nonetheless observed for this variable with high recall frequency being associated with more frequent recording of dreams. This relationship also appeared for the frequency of dreams affecting daytime mood and providing solutions, and with the frequency of déjà vu experiences and reading about dreams.

## Discussion

This investigation described specific dream characteristics and beliefs in a large cohort of French students. By using a validated French version of the MADRE questionnaire, we were able to extend previous findings obtained by surveys using the same comprehensive tool [16,17,20,21] or addressing the same population using non-specialized questionnaires [15].

Interestingly, in our survey we observed important discrepancies between humanities and science students regarding many aspects and beliefs regarding dreams. In our sample, humanities students remembered their dreams and nightmares more frequently, reported more intense and more negative dreams, experienced lucid dreaming more frequently, and had a

more positive attitude towards dreams than science students. As has already been shown, when domains of studies are compared, students differ on personality dimensions with humanities students typically being more open to experiences [14]. By extension, these personality dimensions may also be considered as dominant factors when students choose their future occupational roles. Even if we did not specifically investigate personality dimensions in our cohort, personality differences should be considered as important factors that may explain our findings regarding dream variables, as previously reported in the general population [10,23]. While it has been posited that dreaming may serve to regulate moods [6], our findings indicate that sociodemographic factors may influence this process through complex interrelationships that need to be evaluated before engaging in more specific tests of the function of dreaming. Our results, however, are not in agreement with those of one other study also conducted with French students [15], where no differences were observed between humanities and science students regarding dream recall, nightmares, or lucid dream frequency. This discrepancy could be related to the pre-selection criterion of the former study, where only students without sleep disorders were included, whereas our survey was based on a general voluntary basis without such criteria. In this way, our results may reflect the dream characteristics and beliefs of this very specific population in a more representative fashion. If so, quality of sleep may indeed play an important role as humanities students reported spending more time in bed than science students [15]. However, one limitation of our study is that we did not include a self-assessment questionnaire or any objective measurement of sleep quality. Objective measurement of sleep parameters could better clarify the role of sleep disturbance on dream-related parameters, although such procedures are difficult to apply in large cohorts.

In contrast to what is generally observed [11,16,17,20,21,24], no effect of age was found in the current sample for dream recall frequency, emotional characteristics of dreams, nightmares or lucid dream frequency. When using a different questionnaire, however, this age effect was observed in one study of university students [15]. One possible explanation for this discrepancy may be the rather narrow age range of our sample, as our results were in accordance with those of another study conducted on populations of different ages that showed that the decrease of dream recall frequency only occurs after a mean age of 30 years for men and 40 years for women [25]. Moreover, modifications of sleep characteristics with advancing age and accompanied by decreases in REM sleep stability and total sleep duration, occur later in life compared to the mean age of our sample [26].

The younger age of our sample may also account for the higher dream recall frequency compared to studies of older populations using the MADRE questionnaire [20,21,23]. More specifically, when compared to our previous population [17] where only 13.2% of the sample reported recalling their dreams almost every morning, we observed a higher percentage (18.82%) for this variable in our present sample. Our results are more similar to a Belgian sample [21] that included psychology students and that might have raised the proportion of high recallers by including young students. These proportions are nonetheless different from the ones reported in one study of French students [15], where only 3.00% of their sample reported recalling their dreams every morning. However, the phrasing of the item assessing this variable was different across studies, making any direct comparison difficult [27].

Another important difference between our study and previous investigations was the specific effect of gender in stratified analyses. Considering dream recall frequency, we observed the expected gender effect of women recalling more often their dreams than men, similar to several previous studies of adults and young adults [8,11,12,16,20,21,25,28]. However, we found that women report a higher frequency of dream recall than men independently of their field of study, whereas men in the humanities reported higher dream recall than men in

scientific disciplines. Considering attitude towards dreams, we observed the well-known effect of age whereby older students demonstrated a more negative attitude towards dreams than younger students. However, when stratified by gender, this difference remained significant only for women. Women presented higher scores on attitude towards dreams than men, suggesting that men were less interested in dream content and beliefs than women, but women also tended to lose interest with increasing age. Further evaluation of these differences is warranted, but they should ideally be conducted on samples with equal proportions of men and women. New investigations in adult samples with a wider age span are also needed to evaluate whether this effect of age on attitude towards dreams is specific to women, or if it appears only later in life for men.

In summary, our observation of dream recall frequency in students was consistent with previous studies [16,17,21], and it was a significant predictor of the effect of dreams on waking life. Individuals who more frequently recalled their dreams were more inclined to share their dreams with others or use them to solve problems. Concerning the emotional intensity and tone of dreams among students, as well as the degree of disturbance caused by nightmares or nightmare frequency, our findings appeared to be quite similar to those reported in other adult samples [16,17,21]. When dreams are remembered, they generally tend to be intense and somewhat negative. Women tend to report more intense and more negative dreams than men. A positive relationship between the intensity of dream emotions and dream recall frequency has already been reported [29], and the fact that individuals would better remember dream experiences that incorporate an emotional event has also been investigated in earlier studies [30]. This association may be explained by common cerebral substrates implicated in the processing of memories and emotions during sleep, such as the limbic system and the hippocampus [31,32].

Relative to the tendency to read about dreams, we did not observe differences between the humanities and science disciplines. As hypothesized by a previous study [21], we might have observed a difference if our sample had included more psychology students as such individuals might be more curious about dreaming because of prevalent psychological theories of dreams. Law students constituted the majority of our humanities students, and individuals in this field appear to have no overt reason to be more attentive to their dreams than medical or technology students. By including more students of each discipline, future investigations might be able to elicit specific profiles of dream habits and beliefs.

In interpreting the findings of this investigation, several methodological or conceptual limitations should be considered. First, while the use of a retrospective questionnaire has the advantage of not requiring students to complete a diary every day [27], potential memory biases may have led to an underestimation of the frequency of specific events such as the recall of dreams. Second, a general limitation of studies focusing overtly on dreams is that only people that find this topic interesting may be willing to participate. This may have been the case with individuals participating in the i-Share investigation, and therefore a solution for future investigations may be to evaluate dream characteristics among other questions so as to not announce it as the main topic of the study. Further research is also needed that could focus on other methods and other characteristics of this population in order to establish a more complete portrait of dream recall specificities among students.

## Supporting information

**S1 Dataset. Dataset (Napias et al. 2020).** Our database of 1137 participants.
(XLSX)

**S1 Table. Description of emotional dream variables distribution of the total sample stratified by gender.**
(DOCX)

**S2 Table. Distribution of current nightmares, childhood nightmares and lucid dreams in the total sample stratified by gender.**
(DOCX)

**S3 Table. Distribution of responses for items on question 12 "attitude towards dreams".**
(DOCX)

**S4 Table. Frequency distribution of different effect of dreams on waking life.**
(DOCX)

## Acknowledgments

The authors are grateful to the coordinating team of the i-Share project for their assistance in setting up and collecting the data, in particular Elena Milesi and Clotilde Pollet.

## Author Contributions

**Conceptualization:** Imad Ghorayeb.

**Formal analysis:** Alice Napias, Willy Mayo.

**Methodology:** Imad Ghorayeb.

**Resources:** Emilie Denechere, Imad Ghorayeb.

**Supervision:** Imad Ghorayeb.

**Writing – original draft:** Alice Napias, Imad Ghorayeb.

**Writing – review & editing:** Alice Napias, Willy Mayo, Imad Ghorayeb.

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
