## [Decision Letter · Decision Letter 0]

14 Dec 2020

PONE-D-20-33934

Assessment of dream-related aspects and beliefs in a large cohort of French students using a validated French version of the Mannheim Dream questionnaire

PLOS ONE

Dear Dr. Ghorayeb,

Thank you for submitting your manuscript to PLOS ONE. After careful consideration, we feel that it has merit but does not fully meet PLOS ONE’s publication criteria as it currently stands. Therefore, we invite you to submit a revised version of the manuscript that addresses the points raised during the review process.

I request that you make major revisions before it is processed further. Please carefully consider all issues mentioned in the reviewers' comments. In particular, I reccomend you to focus on these aspects:

re-organization of the introduction, providing an exhaustive context based on the current literature on dreamingclarification of the procedural and methodological aspects (e.g., population description)tables contentslimitations of the studyediting of English language is required

We look forward to receiving your revised manuscript.

Kind regards,

Serena Scarpelli

Academic Editor

PLOS ONE

Journal Requirements:

Reviewers' comments:

Reviewer's Responses to Questions

**Comments to the Author**

1. Is the manuscript technically sound, and do the data support the conclusions?

Reviewer #1: Partly

Reviewer #2: Yes

2. Has the statistical analysis been performed appropriately and rigorously? 

Reviewer #1: Yes

Reviewer #2: Yes

3. Have the authors made all data underlying the findings in their manuscript fully available?

Reviewer #1: No

Reviewer #2: Yes

4. Is the manuscript presented in an intelligible fashion and written in standard English?

Reviewer #1: No

Reviewer #2: Yes

5. Review Comments to the Author

Reviewer #1: The study explored dreams aspects and beliefs in a large sample of French students using a validated online questionnaire (MADRE) to investigate the implication of sociodemographic and cultural variables (age, gender, field of study) in the dreaming process. The results mostly confirm previous work; however, the authors used a validated questionnaire, which is novel.

The work has some appreciable strengths. However, some aspects limit the rigor and potential value of this research.

Please see below for specific details and comments.

-Research questionnaire: given the centrality of the questionnaire on which all results are based, a more detailed description of the measure is needed. It might be helpful to label every point of response scales (here you may find an example: https://doi.org/10.1007/s11818-019-0199-3). Moreover, some items are completely missing in the description (e.g., distress evaluation of nightmares or déjà vu experience).

-Study population: the self-selection recruitment of studies based on online surveys may represent a limit in the interpretation of results, and this should be mentioned in the discussion.

-Population description: I would suggest moving this section in the previous section called “Study population”, for consistency and to improve readability.

-Reporting of the data: the authors report a descriptive table (Table1) of frequency distribution (stratified by gender) only in reference to “Dream recall frequency”. Otherwise, Table 2 present the regression analysis only for “variables concerning dream’s effects on waking life”. I would suggest to upload the descriptive table of the frequency distribution for all items in a supplementary materials section, and adding a summary table for all regression analyses conducted.

-Results: why the authors detailed results considering gender separately only with regard to “dream recall frequency” and “attitude toward dreams”? the authors should clarify the hypotheses on which the analyses are based.

-There seems to be different font sizes used throughout the manuscript (e.g. line 90 and line 114).

-Please ensure that you have this paper language edited and proofread (possibly by a native English speaker) before submitting the revised version.

Reviewer #2: The paper is relevant and of interest for the field but still needs some major revision.

1. Introduction should talk more about what motivated the study. Why is it important to study dreams in a student population? What is the relevance of those finds in the light of the dream theories? Here I suggest looking at the Emotional Regulation Theory from Toren Nielsen.

2. Page 6, line 142: while describing the population, it is crucial to describe the gender distribution in each scientific discipline, comparing the frequencies of male and female. It is not enough to only mention: "Women and men are equally distributed between the two fields of study". As the results are related to gender and scientific disciplines, this issue is essential to be precise.

3. The same population that reveals higher dream recall frequency (women and students from humanities) showed higher intensity in emotional tone. Are these results associated? This would be an interesting topic to explain the global phenomenon in this population. Women at the beginning of their academic careers might have higher academic stress, especially in STEMs. This could impact the emotional tone of their dreams, as well as the dream recall frequency.

4. A better description of what was measured as "attitude toward dreams" before jumping to results can help a better understanding.

5. Sleep disorder should be considered in the results (and even considered as a covariate). Did the authors collected any data related to sleep disorder? If not, this should be mentioned as a limitation.

6. The discussion and the introduction should address more about dreams theories and their relationship with the academic path (pointing out what these results add to the field of dreams).

6. PLOS authors have the option to publish the peer review history of their article (what does this mean?). If published, this will include your full peer review and any attached files.

Reviewer #1: No

Reviewer #2: No

---

## [Author Response · Author response to Decision Letter 0]

28 Jan 2021

All reviewers' comments have been addressed in the attached Response to Reviewers Letter

---

## [Decision Letter · Decision Letter 1]

9 Feb 2021

Assessment of dream-related aspects and beliefs in a large cohort of French students using a validated French version of the Mannheim Dream questionnaire

PONE-D-20-33934R1

Dear Dr. Ghorayeb,

We’re pleased to inform you that your manuscript has been judged scientifically suitable for publication and will be formally accepted for publication once it meets all outstanding technical requirements.

Kind regards,

Serena Scarpelli

Academic Editor

PLOS ONE

Additional Editor Comments (optional):

Reviewers' comments:

Reviewer's Responses to Questions

**Comments to the Author**

1. If the authors have adequately addressed your comments raised in a previous round of review and you feel that this manuscript is now acceptable for publication, you may indicate that here to bypass the “Comments to the Author” section, enter your conflict of interest statement in the “Confidential to Editor” section, and submit your "Accept" recommendation.

Reviewer #1: All comments have been addressed

Reviewer #2: All comments have been addressed

2. Is the manuscript technically sound, and do the data support the conclusions?

Reviewer #1: Yes

Reviewer #2: Yes

3. Has the statistical analysis been performed appropriately and rigorously? 

Reviewer #1: Yes

Reviewer #2: Yes

4. Have the authors made all data underlying the findings in their manuscript fully available?

Reviewer #1: Yes

Reviewer #2: Yes

5. Is the manuscript presented in an intelligible fashion and written in standard English?

Reviewer #1: Yes

Reviewer #2: Yes

6. Review Comments to the Author

Reviewer #1: (No Response)

Reviewer #2: (No Response)

7. PLOS authors have the option to publish the peer review history of their article (what does this mean?). If published, this will include your full peer review and any attached files.

Reviewer #1: No

Reviewer #2: **Yes: **Natália B Mota

---

## [Editor Report · Acceptance letter]

15 Feb 2021

PONE-D-20-33934R1 

Assessment of dream-related aspects and beliefs in a large cohort of French students using a validated French version of the Mannheim Dream questionnaire 

Dear Dr. Ghorayeb:

I'm pleased to inform you that your manuscript has been deemed suitable for publication in PLOS ONE. Congratulations! Your manuscript is now with our production department. 

Kind regards, 

on behalf of

Dr. Serena Scarpelli 

Academic Editor

PLOS ONE